# Methods for Successful Aging: An Aesthetics-Oriented Perspective Derived from Richard Shusterman’s Somaesthetics

**DOI:** 10.3390/ijerph191811404

**Published:** 2022-09-10

**Authors:** Yi-Huang Shih

**Affiliations:** Department of Early Childhood Education and Care, Minghsin University of Science and Technology, Hsinchu 30401, Taiwan; shih262@gmail.com

**Keywords:** older adults, Richard Shusterman, somaesthetics, successful aging

## Abstract

This study explored Richard Shusterman’s somaesthetics to understand the rationale for his view on enhancing the body experience of older adults and increasing their participation in art; it also examined methods or successful aging to enhance the theoretical foundation for educational gerontology. Accordingly, the research objectives were to (1) analyze the definition of successful aging; (2) clarify the role of body experience and participation in art in promoting successful aging among older adults; (3) explore and discuss Shusterman’s somaesthetics; and (4) explore methods for successful aging derived from Shusterman’s somaesthetics. This study mainly explored educational philosophy by collecting, reading, analyzing, logically reviewing, and interpreting the literature on this topic. During this exploration, methods for successful aging were reviewed. The findings are as follows: (1) shifting focus of successful aging to the bodies of older adults; (2) cultivating the body consciousness of older adults enables them to understand themselves and pursue virtue, happiness, and justice; (3) popular art can be integrated to promote the aesthetic ability of older adults and encourage their physical participation in the aesthetic process; (4) older adult education should cultivate the somaesthetic sensitivity of older adults; (5) older adult education should incorporate the physical training of older adults to help them enhance their self-cultivation and care for their body, cultivate virtue, and live a better life; and (6) older adult education should integrate the body and mind of older adults.

## 1. Introduction

In 2002, the World Health Organization (WHO) published *Active Aging: A Policy Framework*, which defines “active aging” as improving the quality of life in old age and implementing the most appropriate processes to promote health, social participation, and safety. Aging should be a positive process that is integrated into national policies and with cultural values to enable everyone to live a quality and dignified life as an older adult. Population aging is a topic of concern in the international community. In response to the trend of global aging, the United Nations and numerous countries (e.g., Japan, Sweden, and France) have established and enhanced scientific research organizations and institutions that focus on older adults to strengthen their geriatric research. In addition, the International Year of Older Persons advocates a “society for all ages” and emphasizes the importance of active aging. Moreover, in 2002, the WHO announced the active aging policy framework, which asserts that the quality of life of older adults should be improved in three aspects: health, participation, and safety. The concept of active aging has gradually developed from the concepts of successful aging, productive aging, and healthy aging. Among these, successful aging is a core concept of health policies for older adults worldwide [1,2,3].

Rowe and Kahn (1997) defined successful aging as a process comprising three elements: (1) reduced incidence of disease and disability; (2) consistently high levels of cognitive and physical function; and (3) active participation in daily activities [4]. Subsequently, scholars such as Crowther et al. (2002) added a fourth element, positive spirituality, which is the positive influence of religious belief and spirituality on older adults [5]. The aforementioned elements of successful aging can be achieved by adopting Richard Shusterman’s somaesthetics.

The propagation and development of culture depend on the application of individual aesthetic abilities to enhance the richness and quality of human life. In the current era of interdisciplinary integration, the cultural industry can facilitate global economic development. By cultivating their aesthetic ability, older adults can develop their talents for diverse pursuits to ultimately live a happy, fulfilling, and beautiful life based on art and culture and successfully age. Body experience and participation in art play pivotal roles in the physical and psychological development of older adults. The literature [6,7] supports this argument. The body experience of older adults and their regular participation in art can be promoted through Shusterman’s somaesthetics.

Traditional philosophy focuses on the mind but neglects the body. However, the increasing influence of body consciousness in the field of philosophy has led to the development of the concept of somaesthetics. The American neo-pragmatic philosopher and aesthetician Shusterman (1949–) pioneered the view of somaesthetics. Shusterman combined the Greek word “soma” (body) with the English word “aesthetics” to coin the new philosophical and aesthetic term “somaesthetics” and establish a basis for developing a new branch of philosophy. Shusterman contends that people pay attention to the body, and this phenomenon has become a key topic in the cultural field [8,9,10,11].

Although Shusterman considers somaesthetics to be a branch of philosophical aesthetics, he also regards it as a theoretical and practical activity with an interdisciplinary nature. Shusterman used the Greek word “soma” instead of the more common word “body” to emphasize the physical aspect of somaesthetics. In contrast to the word “body”, the word “soma” emphasizes the interconnection of the vitality of the mind and thought in the body [8,9,12,13,14,15], reflecting that Shusterman regards the body and mind as inseparable parts of human culture.

In his book *Pragmatist Aesthetics: Living Beauty, Rethinking Art*, Shusterman (2000a) highlights that several scholars have argued that the mind and body are incompatible in the aesthetic experience [9,11,16]. Shusterman attempts to eliminate this persistent prejudice and adopt a constructive strategy. Based on the integration of physical and mental training, he developed somaesthetics as a pragmatically unified physical and mental aesthetic discipline; somaesthetics intends to restore philosophy as a unique way of life, combine philosophy with life, and emphasize the practice of philosophy in the domains of the body and aesthetics. Numerous aesthetic studies have examined the concept of somaesthetics, and various literary theories based on this concept have been developed. Thus, somaesthetics has become a focus of philosophical research [8,9,13,16].

Second, the implementation of body experience and participation in art is essential for older adults. Through body experience and participation in art, older adults can improve their aesthetic sensibilities. In the educational process for older adults, an aesthetic orientation should not be neglected. In fact, the mental activity of older adults is broad. Older adults require a holistic education. Aesthetic sensibilities can fully engage the mental activity of older adults and cultivate the balanced development of the moral and intellectual facets of an older adults’ life [17,18,19].

The present study explored Shusterman’s view of somaesthetics to understand the rationale for the theory’s promotion of body experience, participation in art, and aesthetic cultivation for older adults; it also explored methods for successful aging to enhance the theoretical foundation of educational gerontology. Accordingly, the research objectives were to (1) analyze the definition of successful aging; (2) clarify the role of body experience and participation in art in promoting successful aging among older adults; (3) explore and discuss Shusterman’s somaesthetics; and (4) explore methods for successful aging derived from Shusterman’s somaesthetics.

## 2. Research Methods

Through a theoretical analysis, the present study explored educational philosophy by collecting, reading, analyzing, logically reviewing, and interpreting the literature on this topic [20,21]. First, the researcher directly studied Shusterman’s original work. Subsequently, the researcher consulted the Education Resources Information Center and Scopus databases, and conducted a narrative review of the literature. In other words, the researcher applied the theoretical analysis method, and deductive and inductive reasoning to analyze various arguments and events [20,21]. The researcher examined studies related to Shusterman’s somaesthetics and successful aging, and explored Shusterman’s theory on somaesthetics to understand the reasoning behind his view on enhancing the body experience of older adults and increasing their participation in art; the researcher also examined methods for successful aging to enhance the theoretical foundation of educational gerontology. The conceptual framework of the present study is presented in Figure 1:

## 3. Definition of Successful Aging

According to the U.S. Census Bureau, the global population was projected to almost double from 1980 to 2050 (U.S. Census Bureau, 2004–2005), and the Asia Pacific region will have the largest population of people aged 65 and older in the next two decades. In response to the global issues and challenges related to rapid population aging, researchers are carefully examining the physiological, psychological, and social challenges of old age. Research concepts such as productive aging, successful aging, healthy aging, active aging, robust aging, and gerotranscendence have become focal, and among the aforementioned terms, successful aging was the first to be proposed and is the most frequently cited [2,22].

Wolfe (1990) emphasized that successful aging should be assessed on the basis of physical and mental health, centered on physical and psychological aspects, and less focused on the social aspect [23]; in contrast, the concept of successful aging proposed by Griffith (2001) emphasizes the social aspect and the extent of individual participation in activities without mentioning the physical and psychological aspects [24]. In addition, Rowe and Kahn (1998) emphasized the key roles of the physical, psychological, and social aspects [2].

Rowe and Kahn (1997) published *Successful Aging*, which advocates that “oldness” does not equate to “sickness”. They distinguished between successful aging, usual aging, and pathological aging [4], and numerous studies and theories have considerably contributed to our understanding of aging. Successful aging has become the core of health policies for older adults worldwide [3,22].

Successful aging comprises physical, psychological, and social aspects. The physical aspect involves maintaining good health and an independent life, and the psychological aspect involves adapting well and maintaining normal cognitive function without depression symptoms. The crucial elements are self-training and self-adjustment. When an individual approaches old age, physical and mental activities become more challenging, and they feel that they should relax instead. However, when an individual relaxes, their mind stagnates and they lose their direction in life; consequently, they age faster. Maintaining a high level of cognitive activity is conducive to successful aging [3,4,5,22,25]. Cognitive health (i.e., the ability to clearly think, learn, and remember) is a key component of an individual’s ability to perform everyday activities, and it is also an aspect of overall brain health [26].

Older adults can engage in mental activities (workplace or daily-life experiences) to ensure cognitive retention; this strategy reduces their aging-related cognitive symptoms and provides them with more cognitive resources for managing their cognitive symptoms. The social aspect dimension involves maintaining good family and social relations so that the body and mind are in their optimal state and aging becomes an enjoyable experience. Furthermore, older adults can influence their communities by sharing their life experiences with others, participating in social welfare activities (e.g., helping disabled older adults in the community), and assisting in the preservation of cultural relics and local history to help younger generations understand the history and culture of their living environment. The aforementioned activities can enrich the postretirement lives of older adults. Successful aging is thus only achieved when an individual’s body and mind are healthy and they can enjoy their life [3,4,5,22,25].

## 4. Body Experience and Participation in Art Can Promote Successful Aging

The body experience contributes to successful aging, and the most developed domain of successful aging is physical functioning. Maintaining one’s physical function is a key component of successful aging. Regular physical activity is a major predictor of healthy aging. Decreased muscle mass and muscle strength are not only related to aging processes but also to chronic diseases and health conditions. Therefore, older adults should acquire more body experience through activities such as dance, yoga, qigong, meditation, and tai chi [3,9,27,28].

A study indicated that older adults can develop a positive perceived health-related quality of life by consistently or increasingly engaging in productive body experience and physical activity, especially when they are coping with a life event [29]. Body experience and exercise can help older adults maintain a healthy and active lifestyle [30]. Kanning and Schlicht (2008) indicated that people focus on physical activity because it is a crucial behavior. Physical activity can support successful aging in two respects: it can (1) produce positive physiological and cognitive effects and (2) help older adults to enhance their subjective well-being while they are physically active [31].

Studies have indicated that participation in art contributes to successful aging. On a physiological level, artistic activity contributes to the development of neurons in the brains of older adults [32]. Regular artistic activity is also easy to maintain, and similar to long-term exercise, continued participation in artistic activities challenges the brain to stay engaged. In addition, artistic activities can help older adults to achieve a more relaxed physiological state. At the psychological level, artistic activities can reduce the depression, loneliness, and anxiety of older adults, increase their positive feelings about themselves, and help them to express their emotions. At the social level, artistic activities can increase the social interaction of older adults and increase their life satisfaction [6,32] (Britt-Maj Wikström, 2004; Cohen, 2006). The arts contribute to the health and quality of life of older adults [33].

In addition, a growing body of evidence indicates that active arts engagement can enhance the health and experienced well-being of older adults. Arts engagement can result in participants experiencing (1) positive feelings; (2) personal and artistic growth; and (3) increased meaningful social interactions. Art-based practices promote the well-being of older adults and increase their quality of life [34,35].

Bagan (2022) highlighted the effectiveness of incorporating expressive arts into programs for older adults and patients diagnosed with Alzheimer disease, Parkinson’s disease, or other chronic degenerative diseases [36]. Recent clinical findings validate the long-established consensus among some professionals and people who work with older adults that creating art is an essential activity that provides various health benefits [36] (Bagan, 2022). Several studies have demonstrated that art can reduce the depression and anxiety associated with chronic diseases. Other studies have demonstrated that imagination and creativity can flourish in later life, helping older adults realize their unique potential even when they are affected by Alzheimer’s or Parkinson’s disease [36].

The book *Successful Aging* (Rowe & Robert, 1997) discusses three aspects of successful aging, namely low risk of disease, high mental and physical functioning, and active engagement in life [4]. Expressive art activities can help promote active engagement in life, and art helps older adults to stay engaged in life through positive, healthy, and fulfilling activities [4,36]. The term successful aging is used by the people who work with older adults. Research findings suggest that active and meaningful participation in social life is the primary factor that promotes successful aging; in this respect, the visual arts may play a key role [4,36].

## 5. Shusterman’s Somaesthetics

### 5.1. Connotations of “Soma”

#### 5.1.1. Use of “Soma” to Describe a Sensitive Body Full of Life and Emotions

The term “soma” is used to describe a sensitive body that is full of life and emotions (as opposed to a purely physical body without life and feelings). The body is a crucial and fundamental dimension of one’s identity. It provides the initial perspective through which an individual perceives the world and forms the mode of one’s integration with the world. Often, an individual’s body unconsciously shapes their needs, interests, pleasures, and the abilities through which they achieve their goals [9,10,11].

#### 5.1.2. The Body as the Basic Medium for Human–Environment Interactions

Even if the body is regarded as an instrument, it cannot be neglected. Even as a tool of the self, it is still the most crucial tool among the multiple tools of the self, the basic medium for interacting with various environments, and an essential medium for the development of perceptions, behaviors, and thoughts. Individuals should increase their physical knowledge to enhance their understanding of and performance in various disciplines and practices [9,10,11].

#### 5.1.3. Actions Are Conducted through the Body

Actions can only be performed through the body; willpower is the ability to perform desired actions, and it is dependent on the efficacy of the body. Even if an individual knows how to and has the desire to act correctly, they cannot achieve their goal if they cannot command their body to implement their desired actions. People are surprised when they cannot perform simple physical tasks because only their knowledge is limited and such failures stem from a lack of body consciousness and body control. This situation affects the health and quality of life of older adults [9,10,11].

#### 5.1.4. Inquiry, Attention, and Improvements Related to the Body as the Core of Philosophy

The Platonic philosophical tradition is augmented by modern Cartesian philosophy and idealism, which has discouraged numerous scholars from accepting the long-standing non-Western perception that the exploration, attention, and improvements related to the body form the core of philosophy. However, an individual cannot exist, think, or act without their body. In particular, when philosophy is recognized as a unique way of life and a form of self-consciousness, self-cultivation, and self-care, the body is the core of philosophy [9,10,11].

### 5.2. Nature of Somaesthetics

#### 5.2.1. Somaesthetics Involves Criticism and Improvement

Shusterman proposed somaesthetics, which treats the body experience and art as the core of philosophy and reestablishes philosophy as the art of life [9]. The concept of somaesthetics involves criticism and improvement. Under this concept, the experiences and application of the body are fundamental to perceptual aesthetic appreciation and creative self-shaping. Thus, somaesthetics encompasses knowledge, works, and disciplines that are related to body care and contribute to our understanding of body care. The central goals of body aesthetics (e.g., self-knowledge, body consciousness, somaesthetic sensitivity, correct action, and happiness) are based on the philosophical value of body aesthetics [9,10,11].

#### 5.2.2. Somaesthetics as an Interdisciplinary Theoretical and Practical Activity

Somaesthetics focuses on the body and it can be used to conduct a critical study to examine how people experience their bodies and how they can improve and cultivate their bodies. Thus, somaesthetics encompasses both theory and practice. As a new interdisciplinary field rooted in philosophical theory, somaesthetics provides an integrative conceptual framework and various methodologies that enhance our understanding of our somatic experience and improve the quality of our bodily perception, performance, and presentation. This increased somatic awareness and mastery provides benefits in various fields (e.g., design). Our experience of ourselves and our world involves somatic responses and feelings that are typically unnoticed, even though they are essential to our proficient functioning. We require a proper feel of our tools to use them optimally; this includes the use of one’s body when one is using other tools. The body is an indispensable tool and the necessary medium of our being, perception, action, and self-presentation. Somaesthetics enables us to explore the fundamental features of our embodied methods for engaging with the world and transforming it through action and construction; thus, it can provide useful insights and experiential skills that help designers to create products and situations that lead to rewarding and pleasurable experiences [9,11,37].

#### 5.2.3. Aesthetic Experience from the Perspective of Somaesthetics

##### Life as Aesthetics

The distance between aesthetics and daily life has been closed, and the boundaries between elite and popular cultures have been removed. Art and aesthetics have become crucial components of our daily lives; when life is treated as a form of aesthetics, the aesthetic experience of each person is unique including their aesthetic experience of their body [8,38].

Shusterman (2000b) explores the various roles, methods, and meanings of aesthetic experience in his book *Performing Live: Aesthetic Alternatives for the Ends of Art* [8]. Traditional aesthetic margins reveal the value of aesthetic experience. The aesthetic experience can be achieved through various art forms such as mass media art (e.g., music and movies), the somatic arts of self-cultivation, and the extensive self-stylization art to which body art belongs. These lively art forms are deeply integrated in the pursuit of a superior life as the most advanced form of art (as advocated by scholars) [8].

##### Aesthetic Experience Restores Vivid, Touching, and Shared Experiences That Individuals Seek in Art

People enjoy watching art performances (live or recorded) and obtaining vivid and touching aesthetic experiences through these performances [8]. Shusterman (2000b) indicated that the aesthetic experience can play the role of an “empathy box” that restores the vivid, touching, and shared experiences that a person once sought from art. This phenomenon appears to be a form of nostalgia [8].

##### Aesthetic Experience as an Elevated, Meaningful, and Valuable Phenomenological Experience

As a philosophical concept, the aesthetic experience is an elevated, meaningful, and valuable phenomenological experience as well as a concept with multiple meanings; thus, a traditional art form that is declining may not necessarily become extinct [8].

##### Enhancing and Preserving the Body by Increasing Frequency of Aesthetic Experiences

The more frequently older adults engage in aesthetic experiences, the greater the strengthening and preservation effect the aesthetic experiences will have on their bodies. Focusing on the aesthetic experience facilitates its acquisition. A suitable method for focusing one’s attention on the aesthetic experience is to fully realize its importance and richness [8].

##### Dual Function of Aesthetics in Somaesthetics

Shusterman regards life experience as the concrete realization of art. Hence, experience is art and education, and life experience is an educational process and an object of aesthetics. The meaning of life and educational experience can be appreciated and mastered through the criticism of aesthetic experience. Shusterman argues that aesthetics serves a dual function in somaesthetics. Its first function is to emphasize the perceptual function of the body, and its second function is to emphasize the various uses of aesthetics, which include individual self-stylization and the aesthetic appreciation of objects [9,11,39].

### 5.3. Popular Art Provides Considerable Aesthetic Satisfaction

Although some people regard popular art as a degrading, dehumanizing, and unaesthetic form of art [8,9], the primary argument for popular art is that it provides us with considerable aesthetic satisfaction. Therefore, popular art should not be regarded as an artform that is only suitable for those with unrefined taste and uncivilized societies [8,9].

### 5.4. Comments on Shusterman’s View of Somaesthetics

#### 5.4.1. Condemnation of the Body

##### Intense Emotions and Bodily Misbehaviors Affect Attention

Intense emotions and bodily misbehaviors affect one’s attention. Thus, the body prevents people from truly perceiving the thing-in-itself and acquiring true knowledge. These condemnations underline the core of contemporary media criticism, and our body perception media also distort true ontology because of incomplete perception [9,11].

##### Body as a Simple Tool

The foundational perspective of ancient philosophy regarding the body was adopted and accepted by Neoplatonists and later integrated into Christian theology and modern philosophical idealism. Similar to the discourse in Plato’s *Alcibiades I*, which devalues and alienates the body and regards it as a tool, the aforementioned perspective has considerably influenced various cultures. A corollary that follows the aforementioned line of thinking is that the true self must be the mind or soul. Therefore, self-knowledge and self-cultivation are unrelated to body knowledge and body consciousness. A common line of thinking is that the body is only an external tool for self-use that can be easily transformed into a familiar image. Because the body is perceived as a slave or tool of the soul, it is despised in Aristotelian philosophy. Consequently, Aristotelian philosophy reinforced the subordinate status of numerous terms related to the body and despises them [9,10,11].

##### Body as the Cage of the Soul

Body consciousness may be incompatible with the ideas promoted in contemporary philosophy. However, this phenomenon cannot be attributed to the accusations of numerous body advocates that philosophy has always disregarded the body. In fact, although philosophy has consistently prioritized the mind, the body has always played a central role in philosophy. The primary negative perception of the body is the perception that it is the cage of the soul, the source of sin, and the root of depravity. This negative perception is reflected in and reinforced by the prejudices of Idealism. Therefore, Western philosophers have typically ignored the cultivation of the body and regarded it as the cage of the soul [9,10,11].

#### 5.4.2. Comments on Shusterman’s Somaesthetics

In the late 20th century, various historical and temporal reasons caused the popularity of pragmatic philosophy and aesthetics to increase in the United States. This new form of pragmatism was described by Rorty and Shusterman. New pragmatism references the tradition of Dewey pragmatism and absorbs new elements of contemporary Western culture, especially American culture; thus, compared with Dewey pragmatism, new pragmatism has a higher tolerance and explanatory power. The philosophy of new pragmatism reconciles analytic philosophy with deconstruction and Western philosophy with Eastern philosophy. New pragmatism absorbs numerous elements of Eastern philosophy including Confucianism and Taoism. Shusterman also indicated that he was interested in the Chinese work *Huangdi Neijing*; however, he did not further explore this work because he lacked proficiency in the Chinese language. The incorporation of Eastern philosophical concepts into Western philosophical frameworks indicates that Western philosophers are beginning to attach considerable importance to Eastern culture including that of China. However, this blending of philosophical concepts inevitably creates an internal contradiction between neo-pragmatic philosophy and aesthetics. For example, Shusterman’s pragmatic philosophy and aesthetics combine various aspects such as analytical philosophy, pragmatic philosophy, and practical methods, among which numerous areas of incompatibility are evident. Limitations to ideological development have also been observed for these aspects. The Western term somaesthetics contains the Greek word “soma”, which can mean “survival” [10,11,27,40]. Shusterman neologized the term somaesthetics to establish a branch of embodied philosophy that is explicitly open to combining theory and practice; somaesthetics regards the body as a seat of knowledge and a means of engaging with the world [41]. When Shusterman was asked why he did not name his concept “survival aesthetics”, he responded that he did not want his concept to be confused with traditional ontological philosophy and aesthetics. Shusterman’s theory of new pragmatic philosophy and aesthetics, especially his concept of somaesthetics, is imperfect and contains relatively one-sided truths; however, scholars have still fully affirmed this theory. As of 2021, Shusterman remains a widely recognized and leading researcher in American neo-pragmatic philosophy and aesthetics who actively engages in international aesthetics research [10,11,27,40,42].

## 6. Methods for Successful Aging Derived from Shusterman’s Somaesthetics

### 6.1. Shifting Focus of Successful Aging to the Bodies of Older Adults

Shusterman’s somaesthetics allows philosophy to be prescriptive about everyday embodiments. The concept of “I” is the core of life, and the “body” is the concept that is most closely related to the concept to “I”. The core literacy objectives of older adult education include promoting appropriate living habits; promoting the development of physical and mental health; exploring human nature; exploring the meaning of life, the affirmation of self-worth, and effective career planning; and pursuing perfection and a happiness through self-improvement and transcendence. However, under the concept of “I”, education courses for older adults have focused on the development of personality and on human nature, emotions, the meaning of life, and values and behaviors related to “I”. Notably, the body of older adults is rarely mentioned in the aforementioned education curriculum. The body is thus not the current focus of older adults [9,10,11,43].

The body represents the existence of an individual in time and space, and it is concurrently the simplest and most complex cultural topic. Beauty standards are highly influenced by history and culture, and the definition of the ideal body appears to be historically related to the display of body movements. Moreover, the human body is considered as a symbol and the epitome of nature. Therefore, education for older adults should consider the body as not only a teaching theme, but also the essence of an individual. During the implementation of education for older adults, history, culture, and the physical movements of older adults are merged with nature. This method allows for the themes of older adult education to be understood, explained, and explored. Furthermore, the elements of body aesthetics include beauty, harmony, elegance, transformation, imagery, self-cultivation, creativity, imagination, concentration, style, and Zen [9,10,11].

In Shusterman’s somaesthetics, the body is the medium through which older adults implement their daily life actions and the place where the life experience and value of older adults are shaped. Therefore, somaesthetics is based on the body [38,44]. Similarly, the concept of older adult education is based on the bodies of older adults [45], which are sensitive and full of life and emotions. Somaesthetics focuses on the aesthetic experience of the body of older adults and on refining the elements of the body aesthetics of older adults (e.g., beauty, harmony, elegance, self-cultivation, creativity, imagination, concentration, style, and Zen). Somaesthetics should be incorporated into older adult education to help older adults acquire body experience and cultivate harmonious, elegant, focused, and meditative physical and mental temperaments. Through self-cultivation, older adults can display the beauty of their body rhythms and their creativity and imagination, live unique lifestyles, and enrich their body experience. Moreover, the more frequently older adults engage in aesthetic experiences, the greater the strengthening and preservation effect that aesthetic experiences will have on their bodies [9,10,11,45,46].

### 6.2. Cultivating Older Adults’ Body Consciousness Enables Them to Understand Themselves and Pursue Virtue, Happiness, and Justice

Interpreting body aesthetics is a highly meaningful task. People find it difficult to truly master their bodies. Because of their busy lives, older adults often neglect their body. The exploration of the body is a major topic that warrants attention. The body of a person is influenced by nature and culture. Humans live in nature and are in symbiosis with it; they form communities and create laws, beliefs, religions, customs, and habits. The body can be regarded as an aesthetic concept, and acquiring an understanding of body aesthetics enables people to perceive the inner aspects of their bodies and their body consciousness [9,10,11,47].

Body consciousness is a universal feature of human life. Schusterman argues that body consciousness encompasses not only the mind’s consciousness of the body as an object, but also its embodied consciousness. The living body directly connects with the world and experiences it. Through this connection, an older adult can experience their body as a subject and an object simultaneously. The word “body” is generally regarded as the opposite of the word “mind”. Therefore, the body is often used to refer to an item that is unconscious and lifeless. Moreover, the term “carnal” has various negative connotations in Christian culture and is commonly used in the context of body sensuality. Therefore, Shusterman uses the term “soma” to describe a living, sensitive, dynamic, and perceptual body. However, the body of an older adult is also a living, sensitive, dynamic, and perceptual body [9,10,11,48].

The body lies at the core of Shusterman’s somaesthetics. The goal of somaesthetics is to explore the reasons and methods for improving body awareness. The other objectives of somaesthetics are to encourage individuals to improve themselves and to promote philosophy to achieve traditional goals such as gaining knowledge, knowing oneself, and pursuing virtue, happiness, and justice [9]. Therefore, a goal of older adult education should be to cultivate the body consciousness of older adults and liberate their bodies so that they can truly master their bodies, perceive their bodies, and attain body consciousness. Through older adult education, the self can become independent and the bodies of older adults can become revitalized and sensitive. A dynamic perceptual body is more than a cage for the soul. In addition, body experience is an elevated, meaningful, and valuable phenomenological experience that allows older adults to understand themselves and pursue virtue, happiness, and justice [9,10,11,47,48].

### 6.3. Popular Art Can Be Integrated to Promote the Aesthetic Sensibilities of Older Adults and Encourage Their Physical Participation in the Aesthetic Process

The creation and application of art requires the perception, action, and experience of the body. Popular art can induce people to physically participate in the aesthetic process and it can induce an elevated, meaningful, and valuable aesthetic experience. The aesthetic experience restores the vivid, touching, and shared experiences that individuals seek through art. However, critics of popular art have argued that it prompts passive and effortless participation because of its simple and repetitive structure. This effortlessness can easily lead to people becoming overly fatigued and finding it difficult to engage in challenging endeavors. Thus, popular art cannot evoke an aesthetically positive response [8,9,10].

Shusterman provides a convincing rebuttal to the aforementioned argument by using rock music as an example. Rock music is a typical art form that involves music creation, rhythmic movement, dancing, singing, and even screaming. The audience of a rock concert exerts more effort and performs more actions than the audience of a classical music concert. On the basis of this perception, Shusterman contends that the traditional view that aesthetics is nonutilitarian and separated from daily life is inconsistent with reality. He maintains that aesthetics should focus on physical feelings and experiences, which prevent ignorance and encourage thinking [8,9].

On the basis of the aforementioned view, Shusterman established his unique theory of ‘Somaesthetics’ and emphasized the harmonious unification of the body and mind through the aesthetic process. This harmony allows the body to become flexible and sensitive through physical experience, and it enables older adults to acquire real and rich aesthetic experiences. Thus, the favorable guiding effects of popular art on the bodies of older adults should be recognized [10,11,39,49].

The value of human life is often determined by the living environment of people. Because of the constant evolution of human culture, human environments exhibit various styles, express varying levels of beauty, and increase the value of life. Furthermore, high art and popular art are forms of cultural expression. Aesthetic sensibility is the ability to engage in cultural expression. The aesthetic ability to appreciate creativity and experience sound, music, literature, and art is the basis for older adult education. In this regard, aesthetic ability is the foundation of older adult education; it also represents the ability to appreciate and creatively teach older adults various forms of art [10,11,49].

Older adult educators should integrate popular art into older adult education to encourage older adults to physically participate in the aesthetic process, enhance their body perception, and appreciate the aesthetic characteristics of various things in life; popular art also helps older adults to identify the artistic endeavors that are worth pursuing and the other aspects of life that enable them to realize life as an aesthetic experience. Life experience represents a concrete realization of art. Older adult educators should enrich the aesthetic experience of older adults in their daily lives; help them to appreciate various works or performances of popular art; and increase their ability and tendency to pursue vivid, touching, and shared art experiences. Older adult educators can use the “beauty” in their hearts to connect with the “beauty” in the hearts of older adults. Through this strategy, “beauty” resources can be expanded to generate new possibilities related to the lives of older adults. Film, dance, and music are the most common forms of popular art that older adults can learn about in their daily lives. Art activities that provide high-quality participatory art experiences for older adults are conducted at numerous adult day centers, community centers, and senior centers. Art programs are designed to improve the health of older adults, enhance their quality of life, and combat their feelings of isolation [9,10,45,49,50].

### 6.4. Older Adult Education Should Cultivate the Somaesthetic Sensitivity of Older Adults

The increasing amount of information and perceptual stimuli provided through new human technologies have increased the need to cultivate aesthetic sensitivity to detect and respond to the threats and challenges posed by stressful and excessive stimuli in life. People cannot rely only on advanced technical equipment and must cultivate body sensitivity to monitor their body. Medical problems are mostly related to the function and health of body organs. Therefore, medical personnel must remind patients who have monitoring devices implanted in or attached to their bodies to pay attention to whether these devices are causing physical discomfort or exhibiting signs of malfunction [9,10,36].

A commonly encountered situation is the body’s need to adapt (e.g., body posture and habits) in response to new tools and techniques. Thus, the inefficient use of one’s body and a lack of somaesthetic sensitivity may increase the risk of new physical injuries, discomfort, and disabilities. In contrast, cultivating and enhancing the body’s self-awareness and sensitivity can assist with the discovery and treatment of diseases. Long-term computer use can cause numerous physical problems such as eye fatigue, back and cervical spine pain, various tendon injuries, carpal tunnel syndrome, and other types of recurring stress-induced physical discomfort. These physical conditions are typically caused by poor body posture and habits. Through the cultivation of the body’s aesthetic sensitivity and individual physical self-awareness and self-monitoring, the aforementioned symptoms can be detected [9,10,48].

Art should focus on the mind and life as aesthetics promote the individual physical experience and cultivate individual somaesthetic sensitivity. For example, Dewey became an enthusiastic learner and advocate of the Alexander Technique by practicing somatism. The Alexander Technique is a gentle exercise method for correcting an individual’s upper body posture, promoting movement and breathing, and increasing the awareness and perception of physical behaviors and reactions. In his old age, Dewey devoted more than 20 years to explore his body through the aforementioned technique, which enhanced his body consciousness and somaesthetic sensitivity. Therefore, older adult education should cultivate the somaesthetic sensitivity of older adults to enable them to gain self-awareness and monitor their physical conditions during interactions with the external world. Moreover, older adult education should enrich the physical and psychological experiences of older adults. Regarding the somaesthetic sensitivity of older adults, Shusterman argues that two types of somaesthetic sensitivity exist; the first is the body aesthetics related to the perception of external stimuli, and the second is the body aesthetics perceived from within the body [9,10,11]. Older adults can listen to concerts to cultivate their somaesthetic sensitivity. The diverse external sound stimuli produced by various musical instruments during a concert can stimulate the aesthetic feelings of listeners. After listening to a concert, older adults can be asked to share what they felt inside their body [9,10,11,27,36].

In addition, regular physical activity is a key lifestyle factor in maintaining excellent health among older adults and increasing their life expectancy. Dance is an activity that involves coordinating physical movements with music; it also requires brain activation because a dance activity participant must memorize new dance steps. Dance is a musical–kinetic skill that requires the coordination of body movements with rhythmic stimuli, which promotes the development of movement adaptability. Older adults should be encouraged to participate in dance activities, which help them to develop skills that require the coordination of body movements with rhythmic stimuli, promote movement adaptability, and cultivate somaesthetic sensitivity [9,11,27,50,51].

### 6.5. Older Adult Education Should Incorporate the Physical Training of Older Adults to Help Them Enhance Their Self-Cultivation and Care for Their Body, Cultivate Virtue, and Live a Better Life

The body is a positive object and a tool that helps older adults to control their world. On the use of one’s body as a tool for controlling one’s world, Diogenes, the founder of the Cynic school of philosophy, advocated that physical training is an essential method for developing virtue and improving one’s life. Diogenes cited what he regarded as unquestionable evidence to prove how individuals can easily develop virtue through gymnastics training. To practice the physical training that he advocated, Diogenes subjected himself to various physical challenges (e.g., walking barefoot in the snow and being beaten up by drunk revelers) to test and strengthen his body [9,10,11,52].

Socrates exercised his body and maintained it in a healthy state through regular dance and physical training. He stated that the body is valuable for all human activities. Because of the crucial role of the body in various functions, its health should be maximized. Even for mental activities that require minimal physical exertion, major mistakes frequently occur when people are physically ill. Socrates was not the only ancient philosopher who extolled the virtues of excellent physical health and advocated physical training [9,10,11].

The application of physical training as the principal method for achieving philosophical enlightenment and developing virtues forms the core of Asian physical training exercises such as yoga, qigong, meditation, and tai chi. The Japanese philosopher Yuasa Yasuo asserted that the concept of self-cultivation is similar to that of self-care, and this concept is presumed to be the philosophical basis of Eastern thought because true knowledge can only be obtained through a combination of pure theoretical thinking and body cognition or awareness. Thus, older adults must also develop body cognition or body awareness [9,11,27,52].

In summary, the body is the initial tool that older adults use to understand the world. An older adult’s perception of the world involves harmony between the human body and the world. The desires, needs, and habits of older adults are generally unconsciously experienced but affect older adults and shape their spiritual lives. Shusterman acknowledges that physical training is the basic method for achieving philosophical enlightenment. Asian exercises (e.g., yoga, qigong, meditation, and tai chi) are the core of physical training [9,11,27,53].

Older adult education should therefore involve physical training. For physical training, older adults can engage in activities such as yoga, qigong, meditation, and tai chi. Meditation exercises (e.g., deep breathing and meditation) can produce a steady and high-level flow of pleasure, promote fundamental self-transformation, and provide a joyful and calming experience. Through the aforementioned exercises, older adults can maintain their physical health and focus, enhance their self-cultivation and the self-care of their bodies, cultivate virtue, and live better lives. Thus, these exercises can help older adults to live philosophical lives. Aesthetics is related to the idea of beauty, which is a core concept of literacy for current citizens; virtue is the cultural benchmark of a mature society, and the good life is an indicator of national progress. The three aforementioned aspects are incorporated into older adult education [9,50].

### 6.6. Older Adult Education Should Integrate the Body and Mind of Older Adults

Somaesthetics encourages bodily self-mastery without dualism and with the integration of the body and mind. This strategy can strengthen the theory of educational gerontology [54]. In Shusterman’s theory of somaesthetics, attention must be paid to experiential somaesthetics. Furthermore, sublimity, magnificence, and grace are the basic characteristics of aesthetics. A key point is that an aesthetically pleasing subject should be experienced through physical perception. The concept of “perception and subjectivity” that is emphasized by somaesthetics is highly similar to the concept of “oneness of body and mind” in traditional Chinese culture, thus, the body should not be perceived only as a “machine” or “matter” [9,53].

The term “soma” describes a sensitive body full of life and emotion. It is the most basic medium through which older adults interact with various environments. Individual actions are executed through the body. Inquiries, attention, and improvements relating to the body should be the core aspects of philosophy. Furthermore, the body–mind relationship has always been a key aspect of the development of Western philosophy and aesthetics. The emphasis on the soul over the body has long been the mainstream attitude in Western thought including Socrates, Plato, Augustine, Descartes, and other contemporary philosophers [9,10,18,19,55].

Regarding the body–mind relationship, Shusterman contends that the body and mind are closely related and must not be perceived as two separate entities. The term “body and mind” is suitable for expressing the essential consistency between the body and mind. This term retains the practical distinction between the body and mind from the perspective of pragmatism while emphasizing the unity between the body and mind. On the basis of the integration of physical training and spiritual cultivation, Schusterman proposed the pragmatic unifying aesthetic of the body and mind and positive aging for older adults. The concept of positive aging is increasingly being recognized as a strategy for understanding the lives of older adults worldwide. Positive aging encompasses the various methods that older adults adopt to overcome the life challenges associated with aging; the research into positive aging explores how specific methods enable older adults to age more positively relative to other methods [9,10,17,18,19].

The implementation of older adult education for older adults should be based on aesthetic theory, and the promotion of older adult education can enable older adults to apply their perceptual and rational knowledge to perceive the world. Older adult education can also help older adults to dynamically balance their mind and body and acquire diverse life experiences. Shusterman regards somaesthetics as an aesthetic discipline that integrates the body and mind. This discipline encompasses theory and practice and aligns with the current concept of older adult education; furthermore, it requires the integration of spirituality into older adult care [9,10,55,56,57].

On the basis of the dynamic body–mind balance and diverse life experiences that can be acquired through somaesthetics, older adult education must integrate the body and mind so that older adults can increase their vitality, emotional expression, and sensitivity. In summary, the body and mind of older adults must be emphasized [9,10,57,58].

In addition, the Feldenkrais method influenced Shusterman’s opinions regarding the body’s role in aesthetic perception. Shusterman applied this method to improve the movement of the body and the overall functioning of the body and mind. The Feldenkrais method comprises somatic education techniques that are designed to establish an increased awareness of movements. The desired outcome is to enable an individual to become more functional and spatially (i.e., kinesthetically) aware of their movements during their everyday routine. Relative to other forms of alternative therapy, the Feldenkrais method is a newly developed method. Its techniques are based on the student/teacher paradigm instead of the patient/therapist paradigm. The goal of the Feldenkrais method is to enable an individual to perform various activities (e.g., rolling from their back to their side in bed and reaching for, grasping, and drinking out of a cup) through a standardized learning process. Over time, a student of the method begins to delineate and differentiate the subtle nuances of intention and develop a greater awareness of performance. Throughout this process, the student continually closes the gap between what they intend to do and what they can actually achieve. Overall, by increasing their kinesthetic awareness of their actions, an individual can function at a higher level. The Feldenkrais method promotes the development of awareness through verbally cued movements; it also incorporates functional integration to improve the movement of the body, enhance the overall functioning of the body and mind, and realize body–mind integration. When the Feldenkrais method is implemented in older adult education, its goal is to enable an older adult to roll from their back to their side in bed and to reach for, grasp, and drink out of a cup. An older adult can increase their kinesthetic awareness of their actions to improve the movement of their body and the overall functioning of their body and mind and to achieve body–mind integration [9,10,11,59].

## 7. Reflections and Conclusions

### 7.1. Reflections

The concept of lifelong learning has been in vogue since the 1970s. Various agents at both the grassroots and policy levels have championed it. Lifelong learning was initially applied within a narrow scope that comprised primarily recurrent education and adult education. Although numerous scholars still define lifelong learning as adult education, the definition of the concept has been expanding since the 1990s. From a broader perspective, the concept defines a lifelong learning society as one in which every individual, regardless of age, should be motivated and equipped with the skills to engage in learning on a continuing basis throughout life [60].

The participation of older adults in lifelong learning is an essential factor for successful aging. Because of physical, psychological, and social migration, older adults have numerous problems to overcome. For older adults, lifelong learning is not only an activity for adapting to life but an attitude toward life; under the concept of life development tasks, the six adaptative tasks are adapting to physical aging, adapting to losing a job role, adapting to the death of a spouse, adapting to a reduced income, continuing to participate in society, and maintaining favorable interpersonal relationships. To develop the ability and knowledge required for the tasks that are relevant to each life stage, an individual must keep pace with the times to successfully adapt to changes in life, thus lifelong learning is a necessity. For older adults, lifelong learning enriches the meaning of life and serves as a key method for successful aging [61].

Physical education for older adults is crucial in the context of lifelong learning, and the body is not simply a tool or a cage of the soul for older adults. An older adult’s body is full of vigor and emotion, and the body is the basic medium through which they interact with the environment. Furthermore, body experience and participation in art can promote successful aging. Therefore, the present study explored methods for successful aging derived from Shusterman’s somaesthetics.

Finally, the findings of the present study can be further expanded by exploring the methods for successful aging as derived from Shusterman’s somaesthetics to clarify the linkages among various concepts (i.e., among body, mind, art, exercise [physical training], life-long learning, and successful aging). However, the present study is a philosophical inquiry and not an empirical study, which is one of its limitations. Another limitation must also be addressed. The concept of successful aging may not account for the increasing number of older people who are living with chronic health conditions but are still striving to live lives of value.

### 7.2. Conclusions

The development of healthy habits throughout life is crucial and highly dependent on social and cultural contexts. Therefore, educational programs and interventions in the community must be developed and implemented to promote the positive functioning and well-being of older adults. However, older adult education is regarded as the fastest growing branch of adult education in postindustrial countries and a crucial challenge for adult education [52,60].

Population aging is increasing in numerous countries worldwide, and improving the health of older adults is a key task. Life expectancy is increasing, and healthy older adults continue to contribute to their families and communities; consequently, entire communities are becoming more resilient to poverty. Between 2015 and 2050, the proportion of the world’s population aged 60 years and older is projected to almost double from 12% to 22%. By 2020, the population aged 60 years and older outnumbered the population aged less than 5 years. The pace of population aging is increasing, and countries must overcome major challenges to ensure that their health and social care systems can adapt to this demographic shift. In addition, the United Nations General Assembly declared the decade from 2021 to 2030 as the Decade of Healthy Aging and asked the WHO to lead the implementation of relevant policies. The Decade of Healthy Aging is a global collaboration that brings together governments, civil societies, international agencies, professionals, academia, the media, and the private sector to implement 10 years of concerted, catalytic, and collaborative actions designed to foster longer and healthier lives [62]. The findings of the present study contribute to the existing gerontological literature and serve as a harmonizing theory for successful aging [63]. The present study analyzed Shusterman’s somaesthetics and examined its application in methods for successful aging from an aesthetic perspective.

Through the process of strengthening their body experience and increasing their participation in art, older adults can reduce their incidence of disease and disability, maintain a high level of cognitive and physical function, and actively participate in daily activities. The objectives of the present study were to explore Shusterman’s somaesthetics and interpret the application of somaesthetics in methods for successful aging to enhance the theoretical foundation for educational gerontology.

The present study investigated Shusterman’s somaesthetics and then examined its application in methods for successful aging. The present study discussed novel and key perspectives regarding views of progress and action concepts and ideas that are associated with successful aging. On the basis of the findings, the present study proposes the following recommendations for older adult education. First, successful aging should focus on the bodies of older adults. Second, the body consciousness of older adults should be cultivated to enable them to understand themselves and pursue virtue, happiness, and justice. Third, popular art can be integrated to help older adults to develop their aesthetic ability; this strategy can encourage older adults to physically participate in the aesthetic process. Fourth, older adult education should cultivate the somaesthetic sensitivity of older adults. Fifth, older adult education should incorporate physical training to help older adults to improve their self-cultivation and care for their body, cultivate virtue, and live better lives. Finally, older adult education should integrate the body and mind of older adults.

In addition, the researcher of the present study believes that the aforementioned methods for successful aging can expand the concept of educational gerontology and provide tangible future directions for policy and practice.

## Figures and Tables

**Figure 1 ijerph-19-11404-f001:**
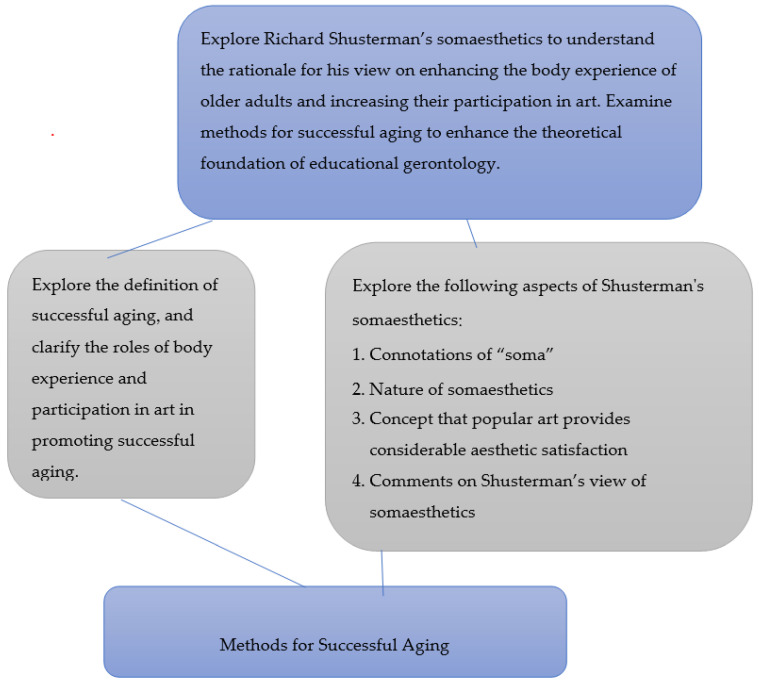
The conceptual framework. Source: Created in the present study.

## Data Availability

All data related to the manuscript are available in the main manuscript.

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
