# Peer review of "Methods for Successful Aging: An Aesthetics-Oriented Perspective Derived from Richard Shusterman’s Somaesthetics"

_ijerph, 2022, doi:10.3390/ijerph191811404_

Round 1
Reviewer 1 Report
Thank you for the opportunity to review the manuscript. The authors present a really interesting discussion with novel and important implications on how we view, progress and action concepts and ideas associated with successful ageing. With that said, this is good discussion paper, however the authors appear to present it as an original study. If this is an evidence review paper with critical discussions presented, then it needs to be structured as such. First, the methods need to be outlined in a separate section. The authors appeared to have conducted a narrative review. Please outline the methodology of a narrative review as it pertains to how this current review was conducted (e.g., what were the key search terms, what were the inclusion criteria and what types of databases were searched). With respect to the findings, the authors introduce a lot of different concepts and this might be overwhelming for some readers not well versed in the subject matter. As such, I would recommend inserting a table with all the key theoretical concepts introduced alongside definitions. This would help to clarify some of the arguments made. For example, the author argues that the 'body' is not of focus on older adult education, but perhaps how the 'body' is defined and perceived in this field may not be aligned to how the author defines body. The body can be perceived as the physical self encompassing the anatomy and physiology of the individual and less in the metaphysical sense. The paper can also be made clearer through better linkages across concepts i.e., how body + mind + art + exercise + successful ageing + life long learning all link together. To me, this is not well characterised. Last, the conclusion can be strengthened with a discussion of the strengths and limitations of the study alongside some clear and perhaps tangible future directions for policy and practice.
Author Response
Author Response Letter
Dear Editor and Reviewer
The author of this manuscript (Manuscript ID: ijerph-1830802) has modified this manuscript according to the reviewers’s comments.
I resubmit this manuscript.
Thank you for the reviewer’s comments.
The authors’ response letter illustrates the revision of this manuscript.
Manuscript ID: ijerph-1830802
|
||||
Methods for Successful Aging: An Aesthetics-Oriented Perspective Derived from Richard Shusterman’s Somaesthetics
|
||||
The first reviewer’s comments
|
Modified page |
Revise
|
||
Thank you for the opportunity to review the manuscript. The authors present a really interesting discussion with novel and important implications on how we view, progress and action concepts and ideas associated with successful ageing. With that said, this is good discussion paper.
|
|
Thanks to the reviewer’s affirmation. |
||
First, the methods need to be outlined in a separate section. The authors appeared to have conducted a narrative review. Please outline the methodology of a narrative review as it pertains to how this current review was conducted (e.g., what were the key search terms, what were the inclusion criteria and what types of databases were searched).
|
3 4 |
Modified according to the reviewer’s opinion. Modify the page as shown on pages 3, 4.
|
||
As such, I would recommend inserting a table with all the key theoretical concepts introduced alongside definitions. This would help to clarify some of the arguments made. For example, the author argues that the 'body' is not of focus on older adult education, but perhaps how the 'body' is defined and perceived in this field may not be aligned to how the author defines body.
|
3
|
Modified according to the reviewer’s opinion. Modify the pages as shown on page 3.
|
||
The body can be perceived as the physical self encompassing the anatomy and physiology of the individual and less in the metaphysical sense. The paper can also be made clearer through better linkages across concepts i.e., how body + mind + art + exercise + successful ageing + life long learning all link together. To me, this is not well characterised. Last, the conclusion can be strengthened with a discussion of the strengths and limitations of the study alongside some clear and perhaps tangible future directions for policy and practice. |
15 16
|
Modified according to the reviewer’s opinion. Modify the page as shown on page 15, 16. Line 750, 751, 752, 753.
|
Thank you for the reviewer’s comments.
Reviewer 2 Report
Thank you for the opportunity to review this interesting manuscript. The manuscript is a very well written piece, which distils a tremendous amount of information on the topic of somaesthetics and aging clearly and coherently, however the method or approach in doing so is not reported and would be beneficial to an appreciation of the findings. On a related matter, there are several unsubstantiated assertions, and whole paragraphs where no evidence is presented in support of the assertions stated, thus in parts, it feels less like an investigation and more like a commentary or perspective piece, albeit founded on logical argument (less so on evidence). Some examples are listed below.
Page 3, lines 128-135, “When an individual approaches old age, physical and mental activities become more 129 challenging, and they feel that they should relax instead.”
This statement (and those following 128-135) seems like a somewhat biased view or a generalisation; if there is evidence to support it, please provide a citation or rephrase it so that it does not appear to be a statement of fact.
Page 4, lines 168 – 174, the narrative around a link between artistic activity and increase in neuronal connections should include citations. The social aspects (only) are cited at the paragraph end.
Page 4, 179-181 requires citations.
Page 4, lines 185-193 refers to “recent clinical findings validate the long-established consensus….” however these are not included (cited).
Page 4, lines 196-198, requires a citation “Expressive art activities can help promote active engagement 196 in life, and art helps older adults to stay engaged in life through positive, healthy, and 197 fulfilling activities.”
Page 4, lines 198-199, “The term successful aging is widely used by the people who work with older adults.” Please consider rephrasing this slightly to omit the word “widely”, since Gerontologists may consider ‘successful aging’ as too narrow a perspective and that it ignores self-perceptions of one’s own aging, regardless of their objective health status, as an important indicator of wellbeing.
Page 7, lines 312-314, “Although some people regard popular art as a degrading, dehumanizing, and unaesthetic form of art, …” this statement should be referenced, it is unclear if it is related to the reference to Shusterman (2000b, 2008) at the latter part of the paragraph.
A minor typo is found at page 4, line 156, “should be acquire more body…”
Page 4, line 173, the word “the” at end of the sentence is extraneous
Finally, it is this reviewer’s recommendation to clearly state the type of publication and to briefly outline the method used to collate the ‘findings’. In addition, consider whether there are limitations that need to be addressed, for example, “successful aging” may overlook the growing number of older people who are living with chronic health conditions, yet are still striving for a valued life?
Author Response
Author Response Letter
Dear Editor and Reviewer
The author of this manuscript (Manuscript ID: ijerph-1830802) has modified this manuscript according to the reviewers’s comments.
I resubmit this manuscript.
Thank you for the reviewer’s comments.
The authors’ response letter illustrates the revision of this manuscript.
Manuscript ID: ijerph-1830802
|
||||
Methods for Successful Aging: An Aesthetics-Oriented Perspective Derived from Richard Shusterman’s Somaesthetics
|
||||
The second reviewer’s comments
|
Modified page |
Revise
|
||
Thank you for the opportunity to review this interesting manuscript. The manuscript is a very well written piece, which distils a tremendous amount of information on the topic of somaesthetics and aging clearly and coherently,
|
|
Thanks to the reviewer’s affirmation. |
||
however the method or approach in doing so is not reported and would be beneficial to an appreciation of the findings. On a related matter, there are several unsubstantiated assertions, and whole paragraphs where no evidence is presented in support of the assertions stated, thus in parts, it feels less like an investigation and more like a commentary or perspective piece, albeit founded on logical argument (less so on evidence).
|
16
|
Modified according to the reviewer’s opinion. Modify the page as shown on page 16. Line 753, 754.
|
||
Page 3, lines 128-135, “When an individual approaches old age, physical and mental activities become more challenging, and they feel that they should relax instead.” This statement (and those following 128-135) seems like a somewhat biased view or a generalisation; if there is evidence to support it, please provide a citation or rephrase it so that it does not appear to be a statement of fact.
|
4 |
Modified according to the reviewer’s opinion. Modify the page as shown on page 4. Line 168, 169, 170, 171, 172, 173, 174, 175.
|
||
Page 4, lines 168 – 174, the narrative around a link between artistic activity and increase in neuronal connections should include citations. The social aspects (only) are cited at the paragraph end.
|
5 |
Modified according to the reviewer’s opinion. Modify the page as shown on page 5. Line 211, 212.
|
||
Page 4, 179-181 requires citations.
|
5 |
Modified according to the reviewer’s opinion. Modify the page as shown on page 5. Line 231, 232.
|
||
Page 4, lines 185-193 refers to “recent clinical findings validate the long-established consensus….” however these are not included (cited).
|
5 |
Modified according to the reviewer’s opinion. Modify the page as shown on page 5. Line 211, 212.
|
||
Page 4, lines 196-198, requires a citation “Expressive art activities can help promote active engagement 196 in life, and art helps older adults to stay engaged in life through positive, healthy, and 197 fulfilling activities.”
|
5
|
Modified according to the reviewer’s opinion. Modify the pages as shown on page 5. Line 230, 231, 232.
|
||
Page 4, lines 198-199, “The term successful aging is widely used by the people who work with older adults.” Please consider rephrasing this slightly to omit the word “widely”, since Gerontologists may consider ‘successful aging’ as too narrow a perspective and that it ignores self-perceptions of one’s own aging, regardless of their objective health status, as an important indicator of wellbeing.
|
5
|
Modified according to the reviewer’s opinion. Modify the pages as shown on page 5. Line 242, 243, 244.
|
||
Page 7, lines 312-314, “Although some people regard popular art as a degrading, dehumanizing, and unaesthetic form of art, …” this statement should be referenced, it is unclear if it is related to the reference to Shusterman (2000b, 2008) at the latter part of the paragraph.
|
8
|
Modified according to the reviewer’s opinion. Modify the pages as shown on page 8. Line 356, 357.
|
||
A minor typo is found at page 4, line 156, “should be acquire more body…” |
5
|
Modified according to the reviewer’s opinion. Modify the pages as shown on page 5. Line 198.
|
||
Finally, it is this reviewer’s recommendation to clearly state the type of publication and to briefly outline the method used to collate the ‘findings’. In addition, consider whether there are limitations that need to be addressed, for example, “successful aging” may overlook the growing number of older people who are living with chronic health conditions, yet are still striving for a valued life?
|
16
|
Modified according to the reviewer’s opinion. Modify the pages as shown on page 16. Line 753, 754,756, 757.
|
Thank you for the reviewer’s comments.
Round 2
Reviewer 1 Report
Thank you for the revisions. I find these to be sufficient for publication.